# Concordant and Discordant Cerebrospinal Fluid and Plasma Cytokine and Chemokine Responses in Mild Cognitive Impairment and Early-Stage Alzheimer’s Disease

**DOI:** 10.3390/biomedicines11092394

**Published:** 2023-08-27

**Authors:** Suzanne M. de la Monte, Ming Tong, Andrew J. Hapel

**Affiliations:** 1Departments of Pathology (Neuropathology), Neurology, and Neurosurgery, Rhode Island Hospital, The Alpert Medical School of Brown University, Providence, RI 02903, USA; 2Department of Medicine, Rhode Island Hospital, The Alpert Medical School of Brown University, Providence, RI 02903, USA; mtong216@gmail.com; 3Department of Genome Biology, John Curtin School of Medical Research, Australian National University, Canberra 2601, Australia; ajhapel@netspeed.com.au

**Keywords:** Alzheimer’s, lumbar spinal fluid, plasma, cytokines, cognitive dysfunction, inflammation

## Abstract

Neuroinflammation may be a pathogenic mediator and biomarker of neurodegeneration at the boundary between mild cognitive impairment (MCI) and early-stage Alzheimer’s disease (AD). Whether neuroinflammatory processes are endogenous to the central nervous system (CNS) or originate from systemic (peripheral blood) sources could impact strategies for therapeutic intervention. To address this issue, we measured cytokine and chemokine immunoreactivities in simultaneously obtained lumbar puncture cerebrospinal fluid (CSF) and serum samples from 39 patients including 18 with MCI or early AD and 21 normal controls using a 27-plex XMAP bead-based enzyme-linked immunosorbent assay (ELISA). The MCI/AD combined group had significant (*p* < 0.05 or better) or statistically trend-wise (0.05 ≤ *p* ≤ 0.10) concordant increases in CSF and serum IL-4, IL-5, IL-9, IL-13, and TNF-α and reductions in GM-CSF, b-FGF, IL-6, IP-10, and MCP-1; CSF-only increases in IFN-y and IL-7 and reductions in VEGF and IL-12p70; serum-only increases in IL-1β, MIP-1α, and eotaxin and reductions in G-CSF, IL-2, IL-8 and IL-15; and discordant CSF–serum responses with reduced CSF and increased serum PDGF-bb, IL-17a, and RANTES. The results demonstrate simultaneously parallel mixed but modestly greater pro-inflammatory compared to anti-inflammatory or neuroprotective responses in CSF and serum. In addition, the findings show evidence that several cytokines and chemokines are selectively altered in MCI/AD CSF, likely corresponding to distinct neuroinflammatory responses unrelated to systemic pathologies. The aggregate results suggest that early management of MCI/AD neuroinflammation should include both anti-inflammatory and pro-neuroprotective strategies to help prevent disease progression.

## 1. Introduction

Alzheimer’s disease (AD), the most prevalent form of dementia-associated neurodegeneration, is marked by progressive behavioral disturbances, impairments in short-term memory and executive functions, and overall deterioration of cognitive faculties [1]. The process begins in a clinically silent or hushed pre-symptomatic stage [2]. The gradual emergence of subtle, early, but concerning declines in recent memory and executive and organizational functions marks the onset of mild cognitive impairment (MCI). The pre-symptomatic and early MCI stages offer the best opportunities to retard or reverse the dementia-bound trajectory. Therefore, strategies to enhance early diagnosis and optimize interventions are AD research priorities. Although the list of options continues to grow, the current standard includes assessments of pTau and Aβ brain pathologies by positron emission tomography (PET) neuroimaging [3,4] and cerebrospinal fluid (CSF)/serum biomarker panels [5,6,7]. However, therapeutic measures designed to abate or cure AD by targeting its dominant structural lesions have met little or no success. Meanwhile, the population of aged individuals who are either at-risk or have already manifest AD continues to grow. As this chronic disease pandemic moves toward crisis mode [8], new approaches are needed to pursue other aspects of neurodegeneration that could be targeted, particularly in the early stages of AD, including with the onset of MCI.

The consistent presence of activated astrocytes and microglia in brains with AD-type neurodegeneration, including in the vicinity of pTau and Aβ lesions [9,10], has led to the concept that neuroinflammation has a pathogenic role [11] such that if treated, AD would be remediated [12]. Mechanistically, activated microglial cells and astrocytes elaborate pro-inflammatory cytokines, chemokines, complement, reactive oxygen species (ROS), and reactive nitrogen species (RNS) [13,14,15]. The increased expression of interferon-gamma (IFN-γ), IL-1β, macrophage migration inhibitory factor, and IL-6 corresponding to pro-inflammatory cytokines in microglia and reactive astrocytes in the vicinity of Aβ plaques suggested links between neuroinflammation and Aβ deposition [9,10]. In addition, neuroinflammation is thought to play causal roles in neuronal injury, cholinergic dysfunction, and stress pathway activation via increased ROS and RNS generation [16]. Subsequent degeneration of nerve terminals could compromise the integrity of synaptic connections and contribute to cognitive decline [14]. If a neuroinflammatory response was routinely found in the pre-symptomatic or MCI stages of neurodegeneration, it would be reasonable to propose its role as an enabler or promoter of disease progression, including transitions from normal aging to MCI and from MCI to AD.

Whether the sources of neuroinflammation are extrinsically derived from systemic factors, intrinsic to the central nervous system (CNS), or combinations of both remains under investigation. Elevated pro-inflammatory cytokines in AD serum samples, reflecting systemic inflammation, have been reported [17,18]. The potential for peripheral-blood-derived cytokines crossing the blood–brain barrier [19] provides a mechanism by which systemic inflammatory responses could contribute to or drive neuroinflammation. By the same token, quashing peripheral inflammation could protect against neuroinflammation and attendant neurodegeneration. In support of this notion, peripheral infections or inflammatory states exacerbate the symptoms of cognitive impairment and worsen cognitive decline [20]. Rigorous clinical trials have, however, failed to demonstrate significant preventive or therapeutic benefits of anti-inflammatory agents on cognitive–neurobehavioral functions in MCI or AD [20,21,22]. Potential underlying challenges and barriers to the success of anti-inflammatory approaches to neurodegeneration include the following: (1) CNS and peripheral inflammation may concurrently mediate neuroinflammation and cognitive impairment in MCI and AD, and therefore both aspects must be addressed; (2) the failure to consider the actions of cytokines, both pro- and anti-inflammatory, and the mediators of ROS and RNS may have rendered previous attempts to target neuroinflammation inadequate; and (3) the timing of neuroprotective anti-inflammatory and antioxidant treatments may have been suboptimum.

This study was designed to address the first two challenges by further characterizing the nature and extent of pro- and anti-inflammatory responses simultaneously present in the CNS and peripheral circulation of patients with MCI or early AD who had been subjected to a thorough clinical evaluation. The approach involved measuring cytokine and chemokine immunoreactivities in paired serum and lumbar puncture cerebrospinal fluid (CSF) samples using a commercial 27-plex magnetic-bead-based enzyme-linked immunosorbent assay (ELISA) to compare systemic and CNS inflammatory, anti-inflammatory, and stress responses relative to subjects without neurological disorders or cognitive impairment.

## 2. Subjects and Methodological Approaches

### 2.1. Human Subjects

This human research study was approved by the Lifespan Hospitals Institutional Review Board (IRB) (board reference #413318 45 CFR 46.110). Samples of normal control lumbar puncture CSF and serum were obtained during diagnosis or via donation within a clinical trial or research study. All participants provided written informed consent for the donations and sample storage for future use; the consent forms were approved by the Lifespan/Rhode Island Hospital. Between 2010 and 2016, the MCI and AD participants were regularly evaluated in the Rhode Island Hospital Alzheimer’s Disease and Memory Disorders Center. Their CSF and serum samples were collected as indicated by the Alzheimer’s Disease Neuroimaging Initiative (ADNI) protocol. NINCDS/ADRDA criteria [1,23] were used to make the diagnosis of AD. Consensus criteria [24] were used to make a diagnosis of MCI. In addition, the MCI patients were followed in the Memory Disorders Center and subsequently (post serum and CSF sampling) determined to have evidence of AD.

Control participants had no underlying neurological disorders and presented in the Lifespan Rhode Island Hospital, Miriam Hospital, or Newport Hospital Emergency Department for minor problems such as headache or back pain between October 2014 and December 2015 when their samples were obtained. In order to be included as a control study participant, the individuals had to be: (1) 21 years of age or older and cognitively intact by neurological exam and hospital records review, with negative screens for neoplasia, underlying inflammatory processes, or any other potential disease confounder; (2) able to donate at least 500 µL samples of lumbar fluid CSF and peripheral blood (for serum) each; (3) free of significant CSF abnormalities via all routine diagnostics; and (4) discharged home in association with and event-free visit in the emergency room. The CSF and serum samples, all free of hemoglobin and other pigments, were stored frozen at −80 °C and passed through a 0.45 µM pore filter prior to their use in biochemical assays. Filtration was needed to ensure the samples were free of cellular debris.

### 2.2. Direct Binding Enzyme-Linked Immunosorbent Assay (ELISA)

We used commercial 96-well-format ELISAs to measure CSF and serum levels of amyloid beta (Aβ1-42) and phospho-tau (pTau-307) [25], adhering to the manufacturer’s protocols. The assays were performed for research rather than clinical diagnostic purposes. Secondary antibody conjugated to horseradish peroxidase and the Amplex UltraRed soluble fluorophore were used to detect immunoreactivity. Fluorescence intensity was quantified in a SpectraMax M5 microplate reader (Ex 565 nm/Em 595 nm) (Molecular Devices, Sunnyvale, CA, USA).

### 2.3. Multiplex ELISA

A Bio-Plex Pro™ Human Cytokine 27-plex Assay (Bio-Rad, Hercules, CA, USA) (Table 1) bead-based ELISA was used to measure cytokines, chemokines, and trophic factors. Serum and CSF were analyzed in parallel according to the manufacturer’s protocol. In brief, after incubating the samples in duplicate with antibody-coupled magnetic beads, biotinylated secondary antibodies and streptavidin-phycoerythrin detected the captured antigens. Fluorescence was measured in a MAGPIX (Bio-Rad, Hercules, CA, USA) device, and the software-generated results (Bio-Rad, Hercules, CA, USA) were reported as cytokines/chemokines/trophic factor concentrations (pg/mL) based on standard curves.

### 2.4. Materials and Reagents

All other fine chemicals were purchased from either Sigma-Aldrich (St. Louis, MO, USA) or CalBiochem (Carlsbad, CA, USA). Bicinchoninic acid (BCA) reagents, horseradish peroxidase (HRP)-conjugated secondary antibody, Superblock (TBS), and enzyme-linked immunosorbent assay (ELISA) MaxiSorp 96-well plates were purchased from Thermo Fisher Scientific (Bedford, MA, USA). Amplex Red soluble fluorophore was purchased from Life Technologies (Carlsbad, CA, USA). The Bio-Plex Pro™ Human Cytokine 27-plex Assay was purchased from Bio-Rad (Hercules, CA, USA). The Luminex MAGPIX system was purchased from Luminex Corp. (Austin, TX, USA). The SpectraMax M5 microplate reader was purchased from Molecular Devices Corp. (Sunnyvale, CA, USA).

### 2.5. Statistics

Data were analyzed by one-way repeated measures analysis of variance (ANOVA) and the post hoc Tukey–Kramer multiple comparison test of significance to compare the control, MCI, and AD sample results. Due to similar trends for the MCI and AD groups, those data were combined (pooled) for comparison with controls. Statistical analyses were performed using Graphpad Prism 10.2. Inter-group comparisons between the MCI/AD and control groups were made using a violin plot with superimposed scatterplots of individual results and unpaired two-tailed Student’s *t*-tests with 5% false discovery corrections. The violin plots included quartile and median values along with the highest and lowest data values. The level of statistical significance was set to *p* ≤ 0.05. In addition, results of 0.05 < *p* ≤ 0.10 are shown as they reflect statistical trends.

## 3. Results

### 3.1. Group Participant Features

The subjects’ age, sex, and Mini-Mental State Examination (MMSE) score data are provided in Table 2. The main inter-group differences were the lower mean age of the controls versus the MCI/AD group (*p* < 0.0001) and the lower MMSE scores in the AD group compared with the MCI group (*p* < 0.05).

### 3.2. AD Biomarkers

The results of the Aβ and pTau serum and CSF ELISAs are provided in Figure 1. The main findings were the significant inter-group differences in the serum levels of pTau, serum, and CSF Aβ and the CSF/serum ratios of pTau and Aβ. In addition, a statistical trend was detected for CSF pTau (Figure 1A). In the AD group, the the serum pTau levels were significantly reduced relative to the MCI (*p* = 0.04) and control (*p* = 0.002) groups (Figure 1B), and the mean serum Aβ was reduced in the MCI and AD groups relative to the controls (both *p* < 0.0001) (Figure 1C). In contrast, CSF Aβ was similarly elevated in the MCI and early AD groups relative to the controls (both *p* = 0.01) (Figure 1C). The mean CSF/serum ratios of pTau and Aβ increased progressively from control to MCI to AD, resulting in significantly higher CSF:serum ratios of pTau (*p* = 0.0002) and Aβ (*p* = 0.006) in the AD group relative to the controls without significant differences between the AD and MCI groups (Figure 1D). These findings reflect progressively reduced Aβ and pTau clearance from the brain with increasing neurodegeneration.

### 3.3. Cytokine and Chemokine Responses

The 27 factors analyzed included chemokines, cytokines, and trophic factors characterized as pro-inflammatory, anti-inflammatory, adhesion-related, or pro-angiogenic, with additional designations as promoters of injury, neurodegeneration, or neuroprotection within the CNS (Table 1). Initial analyses of the cytokine, chemokine, and trophic factor levels in serum and CSF by ANOVA with the Tukey–Kramer post hoc test revealed significant inter-group differences but with concordant directional effects of MCI and early AD relative to the controls for over 90% of the factors (Appendix A). Therefore, the analysis and presentation were streamlined by comparing the combined MCI plus early AD (MCI/AD) results to the controls using two-tailed *t*-tests. The serum and CSF graphed results are shown in Figure 2, Figure 3 and Figure 4 and Figure 5, Figure 6 and Figure 7, respectively.

### 3.4. Serum-MCI/AD Effects on Cytokine/Chemokine/Trophic Factor Expression (Figure 2, Figure 3 and Figure 4)

Among the 27 factors measured, 17 (63%) were significantly modulated in the MCI/AD relative to the control serum. For the nine chemokines, all of which were pro-inflammatory, four (44.4%), including eotaxin, MIP-1α, PDGF-bb, and RANTES, were significantly elevated (Figure 4), two (22%), including IP-10 (Figure 3) and MCP-1 (Figure 4), were significantly reduced, and three (33%), including IL-8 (Figure 2), IFN-γ (Figure 3), and MIP-1β (Figure 4), were either trend-wise altered or unchanged in the MCI/AD relative to control groups. Of the fifteen cytokines, twelve (80%) were significantly modulated and three (20%) were unchanged in the MCI/AD relative to control groups. Among the twelve pro-inflammatory factors, five (42%) were significantly increased (IL-1β, IL-5, IL-13, IL-17a, and TNF-α) (Figure 2, Figure 3 and Figure 4), four (33%) were significantly reduced (GM-CSF, IL-2, IL-6, and IL-15) (Figure 2 and Figure 3), and three (25%) were either trend-wise altered or unchanged (IL-9, G-CSF, and IL-12p70) in the MCI/AD relative to control sera. Of the two anti-inflammatory cytokines, IL-4 was elevated while IL-10 was unaffected (Figure 2). The IL-1ra cytokine receptor adhesion molecule was similarly expressed in the MCI/AD and control sera (Figure 3). Therefore, in the serum, the pro- and anti-inflammatory responses in the MCI/AD group were relatively balanced, with ten pro-inflammatory factors increased, nine pro-inflammatory factors decreased, and one anti-inflammatory factor increased, and three factors remained unchanged. In addition, b-FGF was significantly reduced, while VEGF was unchanged, reflecting some degree of angiogenesis inhibition in the MCI/AD group.

### 3.5. CSF-MCI/AD Effects on Cytokine/Chemokine/Trophic Factor Expression (Figure 5, Figure 6 and Figure 7)

In total, 13 of the 27 factors studied (48%) were significantly modulated in the MCI/AD CSF relative to the controls. Among the nine chemokines, PDGF-bb, which confers neuroprotection, was significantly reduced, while pro-inflammatory MCP-1 and IFN-γ were significantly increased. In contrast, the remaining six (67%) pro-inflammatory/pro-injury chemokines exhibited statistical trend increases (IP-10 and TNF-α), reductions (IL12p70,), or detectable differences (eotaxin, MIP-1α, MIP-1β, IL-8, and RANTES) in the MCI/AD group relative to the controls.

Among the fifteen cytokines, six (40%) were significantly altered in the MCI/AD CSF as follows: four cytokines that promote neuroinflammation, injury, or neurodegeneration (IL-4, IL-5, IL-9, and IL-13) were significantly elevated, and one with neuroprotective effects (GM-CSF) and two with pro-injury/pro-inflammatory properties (IL-17a and IL-6) were reduced (Figure 5, Figure 6 and Figure 7). In addition, one pro-injury/pro-inflammatory cytokine (IL-12p70) exhibited a trend-wise reduction, and another (TNF-α) showed a trend-wise increase in the MCI/AD group relative to the controls. The remaining six (G-CSF, IL-1β, IL-2, IL-10, IL-15, and IL-1ra) were similarly expressed in the MCI/AD and control CSF. Among the trophic factors, VEGF (trophic angiogenesis) and b-FGF (trophic neuroprotection) were significantly reduced (Figure 7), and IL-7 (trophic factor, pro-injury) was significantly increased in the MCI/AD group (Figure 5). Altogether, seven of the twenty-four (29.2%) CSF cytokine and chemokine responses in the MCI/AD group reported an increased activation of pro-inflammatory, pro-injury, or an inhibition of neuroprotective responses, five (20.8%) showed evidence of decreased neuro-inflammation, and twelve (50%) were unaltered relative to the controls. Therefore, like the serum, the chemokine/cytokine modulations in the CSF were mixed but with more pro-inflammatory than anti-inflammatory effects together with a prominent decline in angiogenesis factors.

### 3.6. Concordant and Discordant CSF/Serum Responses in MCI/AD

Databar plots were used to compare the mean percentage differences in the CSF and serum cytokine/chemokine/trophic factor levels between the MCI/AD group and the controls. This data-normalizing graphical approach was used to demonstrate concordant versus discordant shifts in cytokine/chemokine expression in the CSF compared with the serum. The data were subdivided into five clusters based on their degrees of CSF–serum concordant directional responses (Figure 8). Cluster #1 included 10 cytokines/chemokines in which the MCI/AD differences from the controls in the CSF and serum were concordantly either increased or decreased and were statistically significant or showed statistical trends (0.05 ≤ *p* ≤ 0.10). Cluster #2 (N = 7) included factors that were significantly or trend-wise increased or decreased in the MCI/AD serum only. Cluster #3 (n = 4) included significant or statistical trend effects in the CSF with nil responses in the serum. Cluster #4 (n = 3) included fully discordant CSF and serum responses in which the directional shifts in cytokine/chemokine expression were opposite and statistically significant or showed a statistical trend. Cluster #5 (n = 3) included the remaining factors that showed no significant effects of MCI/AD in either the CSF or serum.

In cluster #1, five factors, GM-CSF, b-FGF, IL-6, IP-10, and MCP-1 were concordantly reduced and five, IL-4, IL-5, IL-9, IL-13, and TNF-α were concordantly increased in the MCI/AD CSF and serum. Overall, the net responses (60%) favored pro-inflammatory, pro-injury, or pro-neurodegenerative effects in the CNS, given that two neuroprotective factors (GM-CSF and b-FGF) were reduced, and four pro-inflammatory factors were increased both in the CSF and serum compared with the reduced levels of three pro-inflammatory and the increased expression of one neuroprotective factor (IL-4). In contrast, in the periphery, since GM-CSF is pro-inflammatory and b-FGF is angiogenic, the pro-inflammatory and anti-inflammatory responses in the MCI/AD group were proportionally equal.

Cluster #2 showed significant or trend reductions in G-CSF, IL-8, IL-15, and IL-2, and increases in IL-1β, eotaxin, and MIP-1α in the MCI/AD serum but no significant or trend-wise parallel alterations in the CSF. Since all the chemokines and cytokines included in this cluster are pro-inflammatory, with four reduced and three increased, the net response was an inhibition of systemic pro-inflammatory mediators in the MCI/AD group without a comparable effect in the CNS. On the other hand, it is noteworthy that the prominent but not statistically significant reductions in the MCI/AD CSF levels of G-CSF and IL-8 may have aided in decreasing inflammatory injury in the CNS [54].

Cluster #3 included four cytokines/chemokines that were either significantly or trend-wise reduced (VEGF and IL-12p70) or increased (IFN-γ, and IL-7) in the MCI/AD CSF vis-à-vis nil systemic responses in the paired serum samples. The prominent MCI/AD-associated reduction in CSF VEGF likely reflects a reduced level of angiogenesis as well as a pro-neurodegenerative response since VEGF promotes both angiogenesis and neuroprotection [94,95]. The reduction in IL-12p70 was modest, although it reached a statistical trend. Increased CSF levels of IFN-γ and IL-7 report pro-inflammatory, injury, or neurodegeneration in the CNS [26,41,42,68,83,100]. The overall findings in cluster #3 showed a selectively increased CNS without corresponding systemic pro-inflammatory/pro-injury responses in the MCI/AD group.

Cluster #4 showed opposing directional shifts in terms of cytokine/chemokine expression with significant or trend-wise reductions in the CSF PDGF-bb, IL-17a, and RANTES levels and significant increases in the same molecules in the serum. Since PDGF-bb has neuroprotective effects, its downregulation corresponds to a pro-inflammatory/pro-injury state. RANTES is a pro-inflammatory chemokine, and its reduced expression in the CSF would likely have anti-inflammatory effects. However, both PDGF-bb and RANTES have potential neuroprotective actions related to astrocyte activation [26,90,91,101,102,103,104,105], and therefore, like PDGF-bb, reduced CSF RANTES levels could mediate a pro-injury environment. Reduced levels of IL-17a, which is pro-inflammatory, would likely be protective in the CNS. In contrast, the significantly increased levels of PDGF-bb, RANTES, and IL-17a in the serum reflect systemic pro-inflammatory responses.

Cluster #5 showed heterogeneous but mainly modest MCI/AD-related shifts in IL-1ra, IL-10, and MIP-1β levels in the CSF and serum. None of the inter-group differences reached statistical significance or a statistical trend.

## 4. Discussion

### 4.1. Overview

Emerging data highlight important roles for systemic metabolic dysregulation, particularly insulin resistance, in the pathogenesis of AD-associated cognitive impairment and neurodegeneration [2,106,107,108,109,110,111,112,113,114]. Insulin resistance in brains with AD, and peripherally in people with diabetes mellitus, obesity, or non-alcoholic fatty liver disease, is accompanied and exacerbated by inflammation and oxidative stress. Previously, we reported altered blood–brain barrier permeability [115]/, and distinct but overlapping abnormalities in the expression of insulin-pathway proteins in paired MCI/AD CSF and serum samples [116]. The present work addresses subsequent questions about the co-occurrences of CNS and systemic inflammatory responses in MCI/AD and the degrees to which their alterations overlap. Intersecting patterns of neuroinflammatory with systemic inflammatory responses could reflect common mediators or, considering the blood–brain barrier disruption [115], the trafficking of cytokines and chemokines from the periphery to the CNS. On the other hand, distinct patterns of inflammatory modulation would suggest independent triggers or responses to systemic versus CNS insulin resistance [116]. Relatively few studies have co-analyzed paired serum and CSF samples for inflammation in relation to neuropsychiatric symptoms, cognitive decline, or AD using similar multiplex ELISA approaches [18,31].

### 4.2. Subgroup Characteristics

The above questions were addressed by using a commercial 27-plex cytokine and chemokine ELISA to measure inflammatory markers in paired serum and CSF samples from research study participants diagnosed with MCI, early AD, or no underlying neurological disorder. Corresponding with previous reports, the MCI and AD cases had increased pTau and Aβ CSF/serum ratios, reflecting decreased brain clearances [117,118]. Furthermore, the progressive changes from MCI to AD correspond with the view that CNS clearances of both molecules decline with disease progression. However, since the initial survey of the cytokine/chemokine results showed similar responses in the MCI and AD groups, their data were grouped as MCI/AD to simplify the presentation and results analysis.

### 4.3. Overall Cytokine/Chemokine Alterations in MCI/AD

The systemic functions of the 27 factors were categorized as pro-inflammatory (n = 22), anti-inflammatory (n = 2), adhesion (n = 1), or angiogenic (n = 2). In the CNS, the factor functions were categorized as pro-inflammatory/pro-injury/pro-neurodegenerative (n = 18), anti-inflammatory or neuroprotective (n = 7), or angiogenic (n = 2). In the serum, ten of the twenty-two (45.5%) pro-inflammatory cytokines and chemokines were increased, nine (40.9%) were reduced, and three (13.6%) were unaffected, indicating a mixed but slightly greater pro-inflammatory systemic response in the MCI/AD group. In the CSF, among the eighteen pro-inflammatory factors, six (33.3%) were increased, five (27.8%) were reduced, and seven (38.9%) were unchanged in the MCI/AD group relative to the controls. In addition, for the seven anti-inflammatory or neuroprotective factors, one (14.3%) was increased, two (28.6%) were reduced, and four (57.1%) were not significantly or trend-wise modulated in the MCI/AD CSF. Combining pro-inflammatory with anti-inflammatory effects, eight of the twenty-five factors (32%) reported increased neuroinflammatory/injury/neurodegeneration due to combined increases in pro-inflammatory chemokines or cytokines and reductions in neuroprotective factors, and six factors (24%) reported reduced inflammation, as evidenced by the downregulation of five pro-inflammatory factors and the upregulation of one anti-inflammatory factor. Therefore, although mixed, the CSF pro-inflammatory/pro-injury/pro-neurodegenerative responses were more robust than the anti-inflammatory/neuroprotective effects.

### 4.4. Serum Cytokines/Chemokines—Implications of Specific Alterations

Previous studies have produced varied and inconsistent results regarding alterations in specific peripheral inflammatory markers in MCI and AD [35,79]. For example, one study showed higher serum levels of IL-10, IL-1β, IL-2, and IL-4 in MCI compared with healthy controls [58], while others found no correlation between systemic cytokine elevations and AD progression [63], or they determined that inflammation was more of an early rather than a late marker of AD based on the significantly higher levels of IL-10, IL-1β, IL-4 and IL-2 in MCI but not in dementia [45]. A meta-analysis report showed consistently elevated levels of inflammatory biomarkers such as C-reactive protein, IL-1β, IL-2, IL-6, IL-12, IL-18, MCP-1, MCP-3, IL-8, and IP-10 in AD but inconsistent results for MCI other than elevated levels of MCP-1 [43]. Despite these disparate research outcomes, the recurring theme seems to be that alterations in serum cytokines occur early in the course of the disease but often resolve as AD progresses and may no longer be detectable in the advanced stages of AD [35,58,79]. In essence, systemic pro-inflammatory responses in MCI/AD may serve to propagate tissue injury and potentially drive the progression of neurodegeneration via their transport across the blood–brain barrier. The significantly elevated serum levels of IL-1β, IL-4, IL-5, IL-9, IL-13, IL-17a, eotaxin, MIP-1α, PDGF-bb, RANTES, and TNF-α detected herein corresponded with previous reports [35,43,44,51,58,64,92,119,120,121,122], linking the augmented expression of systemic inflammatory markers to early-stage neurodegeneration.

Increased serum IL-1β and TNF-α levels correlate with cognitive impairment [35] and classical AD histopathologic brain lesions [123]. It is believed that IL-1 activation of astrocytes leading to enhanced S100b levels mediates dystrophic changes in neurites, a loss of synaptic integrity, an increased neuronal generation of Aβ, elevations in intracellular calcium levels, and cell death via excitotoxicity [54,55,56,124]. Increased Aβ levels activate microglia and further increase IL-1β and IL-6 levels [10,125]. Therefore, high levels of IL-1β may initiate or exacerbate self-reinforcing cascades that cause progressive injury, degeneration, and death of neurons in MCI/AD [83].

TNF-α also promotes neuronal injury, driving pro-inflammatory cascades that compromise the viability of neurons, synaptic connections, and gene expression [54,64,65]. TNF-α is known to be elevated with neurodegeneration in humans, including those with AD, motor neuron disease, or Parkinson’s disease [123]. The fact that TNF-α, IL-1β, and other pro-inflammatory cytokines can be actively transported across the blood–brain barrier [126,127,128,129,130] suggests that neuroinflammatory and neurodegenerative responses can be mediated by the CNS trafficking of those systemically derived cytokines.

The significantly increased serum levels of IL-4 and IL-5 in the MCI/AD group correspond with findings in a previous study that reported elevated IL-4, IL-5, IL-1β, TNF-α, IFN-γ, G-CSF, and MIF-1b levels in patients with vascular dementia [36]. This comparison is relevant because AD and cerebrovascular pathologies heavily overlap in cases of vascular dementia, yet the AD component most likely mediates cognitive impairment [131].

The elevated mean level of IL-9 in the MCI/AD serum is noteworthy because an increased expression of IL-9 and IL-12p40 mediated by TNF-α correlates with higher rates of MCI-to-AD conversion [73], and it was found to be associated with modestly elevated IL-9 levels in the CSF of asymptomatic patients with neurofibrillary tangle brain pathology [132]. Its additional relevance to AD is marked by the greater abundance of IL-9-producing Aβ-stimulated CD4+ T lymphocytes [74] and an increased expression along with IFN-γ and IP-10 in the hippocampal tissue of 5XFAD mice [75].

Eotaxins are a chemokine subfamily of pro-inflammatory and anti-trophic proteins. Our findings of elevated eotaxin immunoreactivity in the MCI/AD serum corresponds with previous reports showing its increased levels in the serum and CSF of humans with neurodegenerative diseases, including AD [26,32], and in the serum of aged patients with neuropsychiatric symptoms [31]. Like many cytokines, eotaxins can cross the BBB and exert pathophysiologic damage in the CNS [32]. Therefore, despite the controlled levels of eotaxin in the CSF samples, by crossing the blood–brain barrier, eotaxins can exert injurious effects in the CNS and can contribute to impairments in neurogenesis, cognition, and memory [32].

The serum levels of MIP-1α and IL-13 were significantly elevated in the MCI/AD group. The MIP-1α result corresponds with an earlier finding of increased levels of MIP-1 in the CSF of patients with primary progressive aphasias who also had AD biomarker profiles in their CSF [80]. In contrast, previous reports have found no significant abnormalities in IL-13 levels in the serum or CSF of people with MCI or AD [26,45,58], and in a separate study, the levels of IL-13 and other cytokines were either reduced or undetectable in AD sera and CSF [133]. Contradicting those reports, other studies have demonstrated MIP-1α and IL-13 to be elevated in AD peripheral blood [18,80]. The take-away message is that one must interpret inflammatory marker studies by remaining mindful of differences in populations, disease stages, and methodologies. MIP-1α and IL-13 reinforce the actions of pro-inflammatory cytokines/chemokines and function by attracting inflammatory cells, including T cells, B cells, dendritic cells, and monocytes/macrophages [33]. The findings herein support the concept that MCP-1α and IL-13 participate in systemic neuroinflammatory processes in MCI/AD.

PDGF-BB is the glycoprotein ligand of PDGFR-beta involved in BBB integrity and pericyte maintenance [85,134]. Previous reports have demonstrated elevated levels of PDGF-bb in AD plasma [85,86] and in the soluble fraction of brain tissue [87]. Correspondingly, our study also found significantly higher serum levels of PDGF-bb in MCI/AD relative to the controls. Elevated plasma levels of PDGF-BB were found to correlate with aging-associated cerebral white matter hyperintensities detected by neuroimaging [85], suggesting roles in microvascular-related white matter degeneration and blood–brain barrier disruption in AD and possibly vascular dementia.

The concurrent MCI/AD-associated significant or trend-wise reductions in serum GM-CSF, G-CSF, IL-2, IL-6, IL-8, IP-10, IL-12p70, IL-15, and MCP-1 levels and vis-à-vis the abovementioned increases in the levels of pro-inflammatory factors indicate that the systemic cytokine/chemokine responses were mixed. The MCP-1 decline is noteworthy because, in contrast to other pro-inflammatory cytokines/chemokines, e.g., IL-5, IL-13, and IL-17a that were increased and function by activating Th2 helper cells [33,135], MCP-1 activates TH1 cytotoxic T cells [81]. Two potential interpretations are: (1) the systemic inflammation in MCI/AD is mediated by helper rather than cytotoxic T cell activation; and (2) the reduced serum levels of MCP-1 reflect concurrent cytoprotective host responses.

The lower levels of IL-2, IL-6, and IL-12p70 reported herein conflict with earlier reports [35,43,58,92,121,122]. However, evidence that multiple serum and CSF proinflammatory cytokines and chemokines, including IL-2, IL-1β, IL-6, IL-10 and IFN-γ, decline over time and may be undetectable in AD [133] suggests that related topic study outcomes can vary with disease stage and cognitive decline. The fact that many of these research targets “move” make it difficult to compare results across different populations and studies, highlighting the need to better standardize subject characteristics to interpret pro-inflammatory and anti-inflammatory responses along the normal aging to MCI to AD spectrum. The negative results from large AD-focused studies [70,133] reinforce the concept that disease staging is critical for the accurate interpretation of data generated by multiple studies with different designs, particularly since systemic inflammation may represent an initial, early-stage response to mediators of neurodegeneration that evolve or resolve as factors contributing to disease shift over time.

Regarding angiogenesis, serum bFGF was significantly reduced and VEGF was unchanged in MCI/AD. IP-10, which functions as a chemokine with angiostatic properties [41,76], was also reduced in the MCI/AD serum. One consideration is that in MCI/AD, peripheral angiogenesis is suppressed due to the inhibition of bFGF with possible contributions from the reductions in IP-10. Mechanistically, these responses could account for the well-recognized micro-vascular dysfunction and pathology in peripheral-insulin-resistance diseases [136] and AD [136,137,138], which also is associated with brain insulin resistance [136].

### 4.5. CSF Cytokines/Chemokines—Implications of Specific Alterations

In the CSF, the significant or statistically trend-wise increases observed for IL-9, TNF-α, IL-13, IL-5, IFN-γ, and IL-7 mark pro-inflammatory, pro-injury, or pro-neurodegeneration responses in the CNS. The accompanying significant or statistically trend-wise reductions in the levels of GM-CSF, b-FGF, and PDGF-bb reflected reduced neuroprotective mechanisms, compounding the deleterious effects of an enhanced expression of pro-inflammatory/pro-injury cytokines and chemokines. Like the serum, the responses in the CSF were mixed, in that the levels of IL12p70, IP-10, MCP-1, IL-6, and IL-17a, which have pro-inflammatory or pro-injury effects, were significantly or trend-wise reduced, and the level of IL-4 (neuroprotective) was increased. However, unlike the serum, the dominant responses in the CSF were pro-inflammatory, pro-injury, or pro-neurodegenerative. In addition, a significant downregulation of VEGF and PDGF-bb would have served to impair angiogenesis and blood–brain barrier integrity and thereby contribute to AD-associated CNS vascular degeneration [131,139,140,141,142,143,144,145].

The MCI/AD-associated modulation of IL-4, IL-5, IL-7, IL-9, TNF-α, IFN-γ, and VEGF levels in the CSF were largely in agreement with earlier studies, several of which employed similar approaches but with samples that included later-stage AD [26,31,77]. In contrast, discordant results relative to the published literature were obtained for PDGF-bb, MCP-1, IP-10, eotaxin, and IL-10, which were either reduced or not modulated in the CSF from our MCI/AD cases, but they were increased in earlier studies [26,31,77,78]. The variability in these research outcomes could be attributed to differences in the stages of MCI or AD between our study and the published data. For example, with the progression of AD, CNS neuroinflammation tends to wane [133] and therefore could fail to show effects compared with early-stage AD or MCI. In addition, the heterogeneity within the diagnostic categories for MCI and early AD combined with the limited sample sizes led to a large statistical variance, which for some factors may have precluded the detection of true underlying inter-group differences.

The observed MCI/early-AD-associated modulations of seven CSF cytokines and chemokines in favor of increased inflammation, injury, or neurodegeneration, including TNF-α, IFN-γ, IL-5, IL-9, and IL-13, or reduced neuroprotection (PDGF-bb and GM-CSF) correspond with previous reports on AD [146,147]. TNF-α, a promoter of excitotoxic injury, is upregulated or dysregulated in AD, Parkinson’s disease, and motor neuron disease [64,123]. Its increased immunoreactivity around Aβ senile plaques suggests a role in relation to one of the signature AD lesions [64]. In addition, increased levels of TNF-α induce neuronal injury via its capacity to drive pro-inflammatory cascades that challenge neuronal survival, synaptic function, and gene expression [54,64,65]. Since pro-inflammatory cytokines, including TNF-α and IL-1β, are expressed in activated astrocytes and microglia [147] but can also be actively transported across the blood–brain barrier [126,127,128,129,130], the associated neuroinflammatory and neurodegenerative responses could be mediated by pro-inflammatory cytokines of CNS, systemic, or both origins.

The pro-inflammatory role of IFN-γ as a mediator of neurodegeneration has been established [41]. An increased CNS expression of TNF-α and IFN-γ may mediate AD-associated increases in CSF IL-7 levels [46]. IL-5, IL-7, IL-9, and IL-13 levels are all increased in MCI and/or AD and are detectable in CSF [26,48,61,66], as observed herein. IL-7 in the CNS enhances the proliferation of myelin-activated T cells [46], highlighting its potential contribution to white matter degeneration, which begins in the early, pre-symptomatic stages of AD [139,148]. IL-5 neuroinflammatory injury is mediated by the enhanced proliferation and activation of microglia and Th2 helper cells, leading to an increased generation of nitrosative stress [149]. IL-9 promotes the migration of T cells into the CNS [66]. Increased IL-13 levels, like TNF-α, promote cortical excitability and are associated with Aβ deposition in AD [48]. Together, these observations support the concept that elevated levels of cytokines and chemokines that promote neuroinflammation, neuronal injury, or neurodegeneration are features of MCI and early AD and are detectable in intrathecal CSF samples. Furthermore, these findings suggest that optimally timed and targeted anti-inflammatory interventions such as those with anti-IL-7, anti-TNF-α, and anti-IFN-γ biologicals may reduce or prevent neurodegeneration, including white matter, in the early stages of AD. Conceivably, several of the already-developed anti-cytokine and cytokine-receptor-blocking biologics [150,151] could be re-purposed for the early treatment of AD. Although not significantly or trend-wise modulated in the MCI/AD CSF, the significantly elevated levels of systemically-derived IL-1β, eotaxin, and MIP-1α may have contributed to the pro-inflammatory CNS state, since these factors can cross the blood–brain barrier [126,127,128], and the blood–brain barrier was previously shown to be disrupted in MCI/AD cases [115].

Several anti-inflammatory factors were significantly or trend-wise modulated in the MCI/AD CSF. PDGF-bb, GM-CSF, and b-FGF levels, which also have neuroprotective effects, were reduced in the CSF and therefore likely contributed to a pro-inflammatory state within the CNS. PDGF-bb promotes neuronal survival and neurogenesis [82,88,89]. Its significantly reduced levels in the MCI/AD CSF suggest that anti-survival and anti-growth pathways are activated early in neurodegeneration. GM-CSF has neuroprotective actions in the brain, as shown by the reduction in cerebral infarct size [37] and the reduction in neuropathology and cognitive impairment in an AD transgenic mouse model [38]. Similarly, the significantly reduced CSF levels of b-FGF point to impairments in neuroprotection and growth signaling [27,28,96] in MCI/early-stage AD. IL-4 was the only anti-inflammatory or neuroprotective factor that was significantly elevated in the CSF, potentially countering the effects of diminished PDGF-bb, GM-CSF, and b-FBF levels. Nonetheless, the aggregate findings suggest that in MCI/AD, neuroprotective measures were barely in play relative to pro-inflammatory and pro-neurodegenerative responses. On the other hand, previous studies have suggested that the pro-inflammatory CNS status can resolve with AD progression such that pro-inflammatory factors decline and anti-inflammatory factors, including b-FGF, increase or normalize in association with senile plaques, neurofibrillary tangles, and neuropil threads in the later stages of AD [27,29]. Therefore, strategies for rectifying altered cytokine/chemokine profiles in AD as well as other neurodegenerative diseases must consider the potential for host responses to shift over time, such that therapeutic approaches effective at one stage of AD may be ineffective or detrimental at another.

VEGF has major roles in CNS angiogenesis [95,97,98]. In addition, besides its neuroprotective effects, b-FGF also induces angiogenesis [30]. The significantly reduced CSF levels of b-FGF and VEGF point to impairments in neuroprotection, growth signaling [27,28,96], and angiogenesis, which are needed to support micro-vascular perfusion [95,152]. The lower levels of VEGF in the MCI/AD CSF correspond with a previous report of declining VEGF in CSF and brains with neurodegeneration [96]. A reduced VEGF expression correlates with hippocampal atrophy, loss of executive function, and declines in memory [96]. Microvascular pathology with evidence of impaired perfusion and ischemic lesions is a recognized feature of white matter atrophy and degeneration in AD [136,139,148].

### 4.6. Concordant Versus Discordant CSF-Serum Responses

In the CNS, pro-inflammatory cytokines and chemokines can be generated locally by activated microglia and astrocytes [14,26,35,54,125], or they could be transported across the blood–brain barrier from the peripheral circulation [52,153]. Activated microglia and astrocytes can attract the migration of T cells to the CNS, where they release pro-inflammatory cytokines and chemokines [146,154]. In addition, environmental cue-activated neuronal and endothelial cells can generate regional gateways to attract the ingress of pathogenic T cells that cause CNS injury [155]. The analysis of the paired CSF and serum samples provided an opportunity to characterize and distinguish endogenous from exogenous neuroinflammation. The concordance of the inflammatory profiles suggests that the CNS and systemic responses in early-stage neurodegeneration were similar, whereas the discordant aspects imply independent and disparate CSF-serum responses. Whether the sources of neuroinflammation were endogenous to the CNS, exogenous, or both, evidence suggests that the cytokine signatures associated with diseased neurons significantly impact gene expression in otherwise normal brain cells [75] and thereby contribute to AD progression.

Among the 27 factors examined, 10 (37.0%) were significantly or statistically trend-wise concordantly increased or reduced in the serum and CSF. The main effect for six factors was either anti-neuroprotective (reduced GM-CSF and b-FGF levels) or pro-neuroinflammatory (increased IL-5, IL-9, IL-13, and TNF-α levels). This suggests that either the underlying driving factors were shared or systemic factors crossed the blood–brain barrier to promote very similar inflammatory profiles in the CNS and periphery. The same argument could be made for the parallel reductions in the levels of pro-inflammatory mediators (IL-6, IP-10, and MCP-1) and the increase in serum and CSF IL-4 levels, which would have countered the neuroinflammatory responses. The parallel pro- and anti-inflammatory profiles in the CSF and serum favor the concept that the CNS and systemic responses are mediated by similar pathologies. A likely cause is the dysregulation of the CNS and systemic metabolic pathways linked to insulin resistance based on earlier findings in the same samples [116] and the well-established relationship between insulin resistance and inflammation in AD [92,136,156,157]. However, the alternative hypothesis that systemic insulin-resistance diseases including obesity, diabetes mellitus, non-alcoholic fatty liver disease, and metabolic syndrome drive peripheral and CNS inflammatory processes leading to cognitive impairment and neurodegeneration is also plausible and, if proven correct, would provide a feasible target for diagnostics and therapeutic intervention.

Discordant CSF-serum responses were observed for seven cytokines or chemokines. One cluster (#3) showed statistically significant or trend-wise responses in the CSF but nil responses in the serum. Cluster #4 showed completely opposite responses in the CSF and serum. These findings suggest that a subset of the CNS and systemic inflammatory processes were separate and distinct with possibly non-overlapping etiologies. Moreover, the discordant results characterized by increased neuro-inflammation vis-à-vis absent or diminished systemic responses argue in favor of brain-specific neuroinflammatory processes that may contribute to cognitive impairment and neurodegeneration.

### 4.7. Strengths and Limitations of the Study

The main strength of this study was the simultaneous analysis of the serum and CSF obtained from the same individuals at the same time points to evaluate how the systemic (peripheral blood) and neuro-inflammatory profiles were differentially modulated in MCI/AD compared with normal subjects. The approach enabled the characterization of the differential patterns of cytokine/chemokine activation or inhibition at relatively early stages of neurodegeneration. From a mechanistic perspective, a better understanding of the role of inflammation as a mediator of neurodegeneration is needed to make further decisions about the potential utility of early anti-neuroinflammatory versus neuroprotective therapeutics. The findings indicate that neuro-inflammation and systemic inflammatory responses occur concurrently in MCI/AD, but they also suggest that neuro-inflammation in MCI/AD is mechanistically driven by more than a single process. In this regard, the results suggest that some aspects of neuroinflammation may be driven by two or three processes including CNS pro-injury/pro-inflammatory factors, systemic pro-inflammatory factors that cross the blood–brain barrier, and the inhibition of neuroprotective factors. Failure to address CNS-predominant early neuroinflammatory mediators such as IL-7 and IFN-g, and perhaps more important, the lack of neuroprotective and pro-angiogenic strategies could account for prior failures to modify the course of AD with anti-inflammatory agents [22,158]. An additional potential barrier to prior success may have been the limited ability of the therapeutic compounds to cross the blood–brain barrier [159]. Future therapeutic trials should incorporate neuroprotective agents that support or bolster neuronal functions challenged by metabolic dysregulation, the activation of neuroinflammatory chemokines and cytokines, and the suppression of endogenous CNS neuroprotective measures.

A definitive interpretation of the data may be limited by the cross-sectional nature of the study, in which the CSF and serum samples were obtained at a single time point. Ethical considerations precluded repeated longitudinal sampling, particularly of the normal participants. The relatively small group sizes and age differences between the MCI/AD and control groups were additional limitations of the study. However, since seven of the twenty-seven serum and eleven of the twenty-seven CSF factors were not significantly modulated by diagnosis and ten of the twenty-seven factors were discordantly modulated by diagnosis, it is likely that the observed pathophysiological responses were MCI/AD-related rather than strictly age-dependent. The lack of a cytokine polymorphism analysis prevented the identification of genetic factors that may have been responsible for specific pro-inflammatory responses. If feasible and justified, future studies could be designed to characterize longitudinal, age-related versus MCI/AD-associated shifts in systemic versus CNS inflammatory responses to better understand the causes and consequences of neuroinflammation, perhaps in manners that diagnostically and mechanistically distinguish aging from MCI from AD.

## Figures and Tables

**Figure 1 biomedicines-11-02394-f001:**
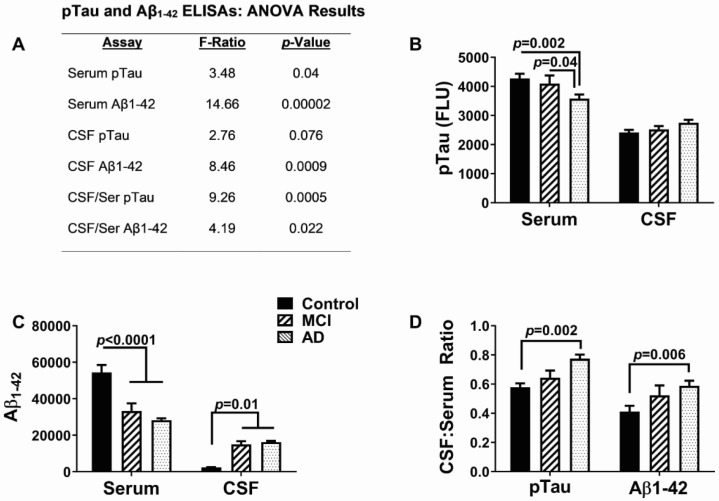
AD clinical biomarkers: The levels of immunoreactivity corresponding to amyloid-beta peptide, 1–42 (Aβ_1–42_), and phospho-tau (pTau-307) were measured in serum and CSF samples from normal control, MCI, and AD study participants by ELISA. Measurements are expressed as fluorescent light units (FLU). (**A**) ANOVA test results (F-ratios and *p*-values). Graphs depicting the mean ± S.D. of (**B**) pTau immunoreactivity, (**C**) Aβ_1–42_ immunoreactivity, and (**D**) the calculated CSF:serum ratios of pTau and Aβ_1–42._ Significant inter-group differences detected by the post hoc Tukey–Kramer multiple comparisons test are shown in the panels.

**Figure 2 biomedicines-11-02394-f002:**
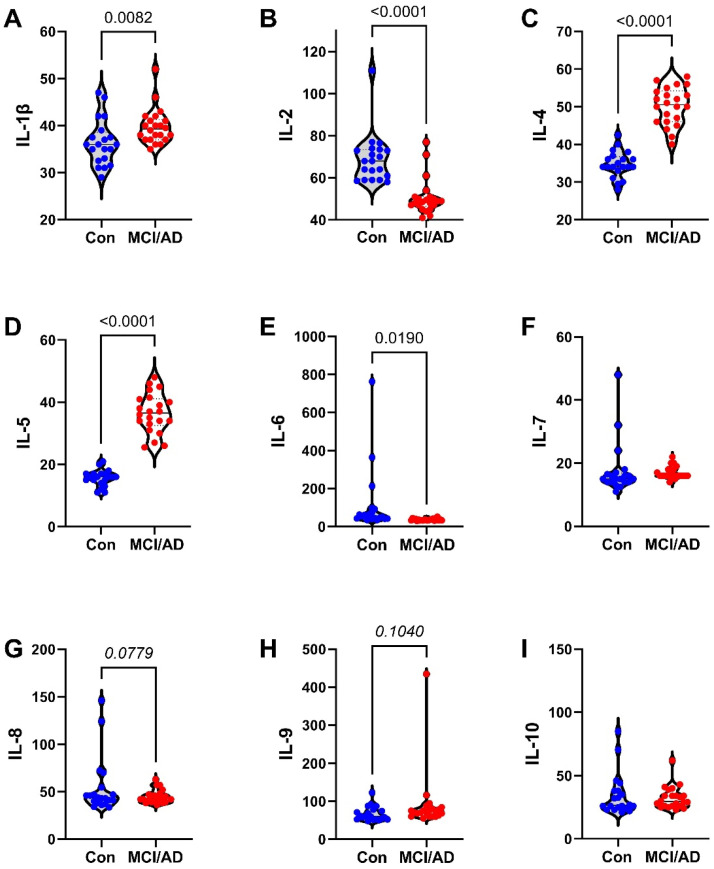
MCI/AD effects on serum cytokine and chemokine expression—1: Bead-based multiplex ELISAs were used to measure cytokines, chemokines, or trophic factors in serum. Violin plots depict inter-group comparisons with individual values corresponding to the control (Con; N = 21) and MCI/AD (N = 18) levels of (**A**) IL-1β, (**B**) IL-2, (**C**) IL-4, (**D**) IL-5, (**E**) IL-6, (**F**) IL-7, (**G**) IL-8, (**H**) IL-9, and (**I**) IL10 analyzed using Student’s *t*-test. Significant (*p* < 0.05) and statistical trend (*0.05* < *p* < *0.10*; *italicized*) inter-group differences are shown within each panel.

**Figure 3 biomedicines-11-02394-f003:**
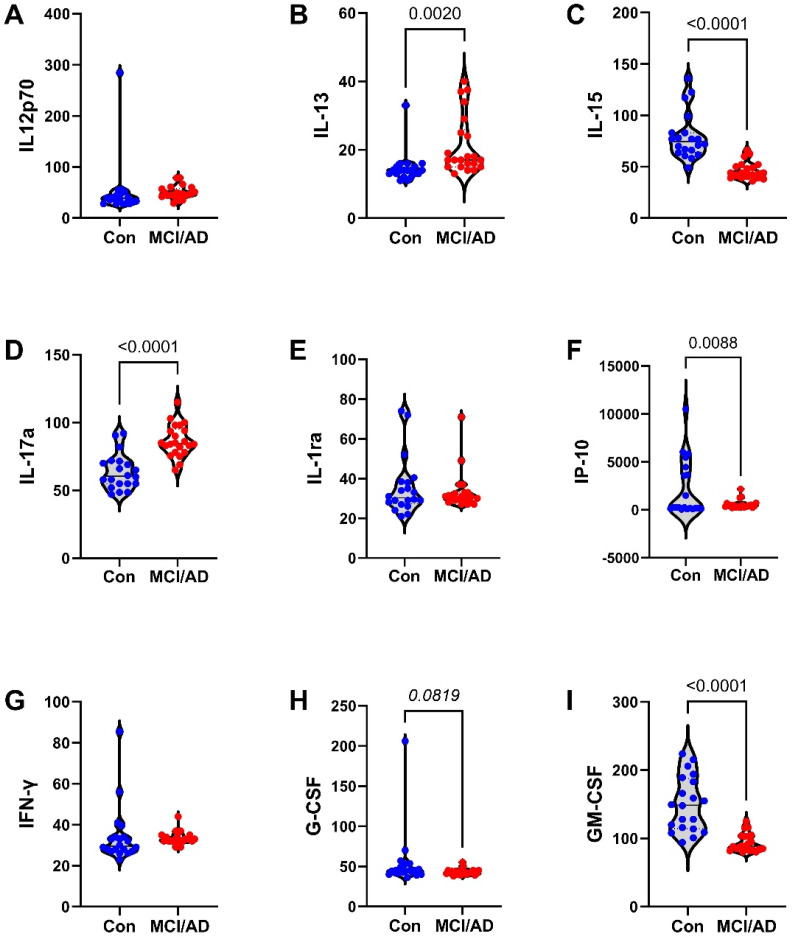
MCI/AD effects on serum cytokine and chemokine expression—2: Bead-based multiplex ELISAs were used to measure cytokines, chemokines, or trophic factors in serum. Violin plots depict inter-group comparisons with individual values corresponding to the control (Con; N = 21) and MCI/AD (N = 18) levels of (**A**) IL-12p70, (**B**) IL-13, (**C**) IL-15, (**D**) IL-17a, (**E**) IL-1ra, (**F**) IP-10, (**G**) IFN-γ, (**H**) G-CSF, and (**I**) GM-CSF analyzed using Student’s *t*-test. Significant (*p* < 0.05) and statistical trend (*0.05* < *p* < *0.10*; *italicized*) inter-group differences are shown within each panel.

**Figure 4 biomedicines-11-02394-f004:**
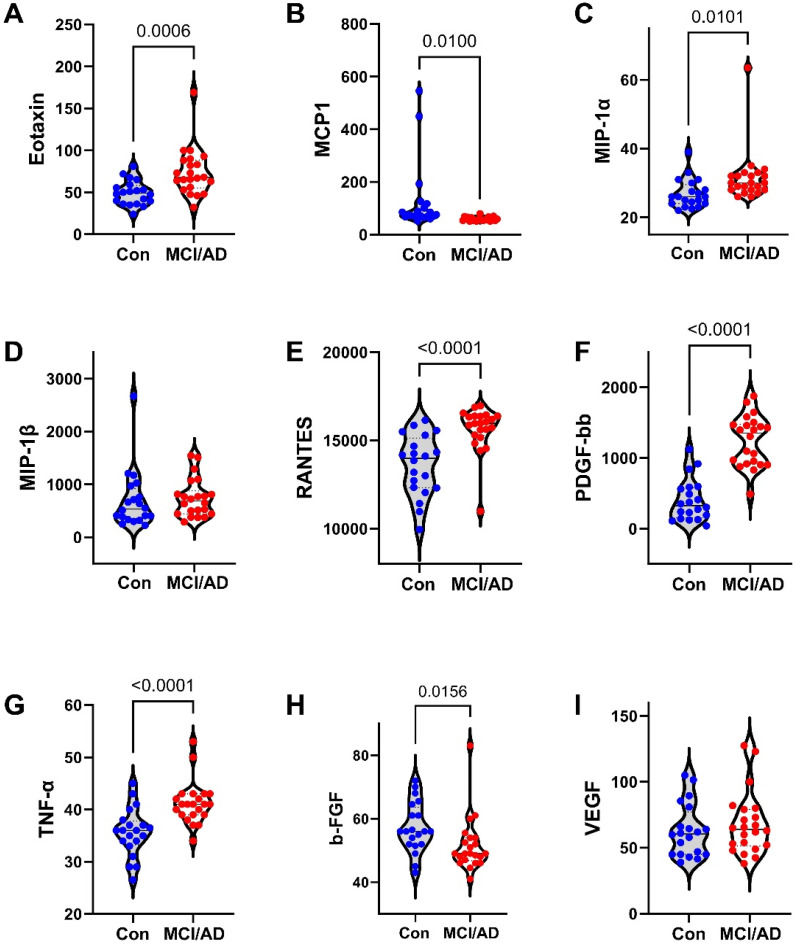
MCI/AD effects on serum cytokine and chemokine expression—3: Bead-based multiplex ELISAs were used to measure cytokines, chemokines, or trophic factors in serum. Violin plots depict inter-group comparisons with individual values corresponding to the control (Con; N = 21) and MCI/AD (N = 18) levels of (**A**) eotaxin, (**B**) MCP-1, (**C**) MIP-1α, (**D**) MIP-1β, (**E**) RANTES, (**F**) PDGF-bb, (**G**) TNF-α, (**H**) b-FGF, and (**I**) VEGF analyzed using Student’s *t*-test. Significant (*p* < 0.05) and statistical trend (*0.05* < *p* < *0.10*; *italicized*) inter-group differences are shown within each panel.

**Figure 5 biomedicines-11-02394-f005:**
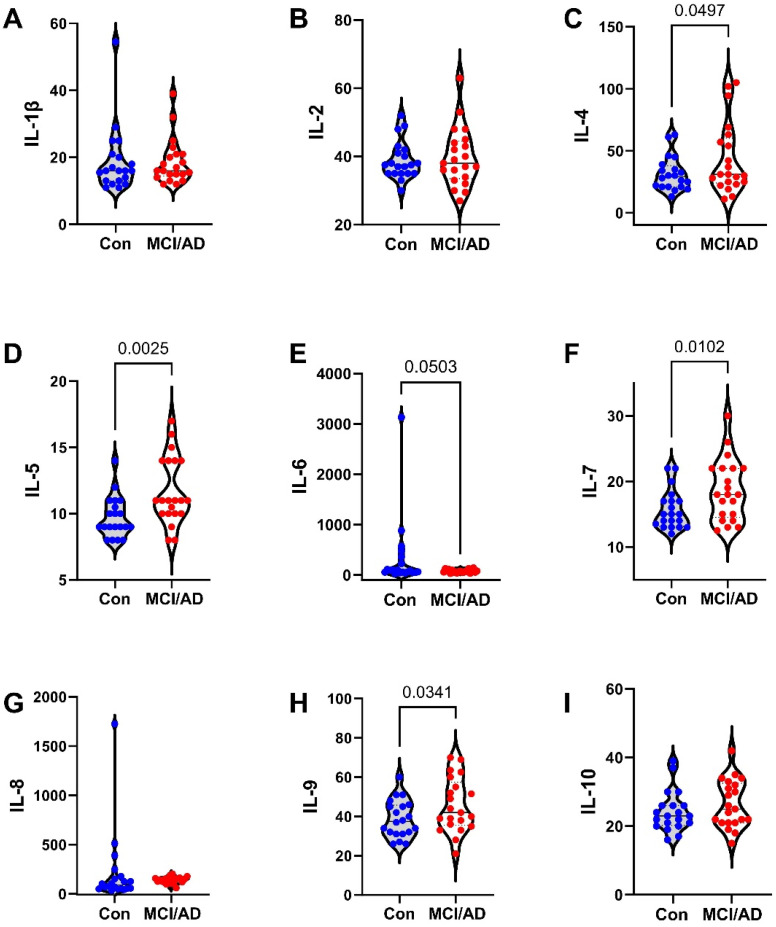
MCI/AD effects on CSF cytokine and chemokine expression—1: Bead-based multiplex ELISAs were used to measure cytokines, chemokines, or trophic factors in CSF. Violin plots depict inter-group comparisons with individual values corresponding to the control (Con; N = 21) and MCI/AD (N = 18) levels of (**A**) IL-1β, (**B**) IL-2, (**C**) IL-4, (**D**) IL-5, (**E**) IL-6, (**F**) IL-7, (**G**) IL-8, (**H**) IL-9, and (**I**) IL10 analyzed using Student’s *t*-test. Significant (*p* < 0.05) and statistical trend (*0.05* ≤ *p* ≤ *0.10*; *italicized*) inter-group differences are shown within each panel.

**Figure 6 biomedicines-11-02394-f006:**
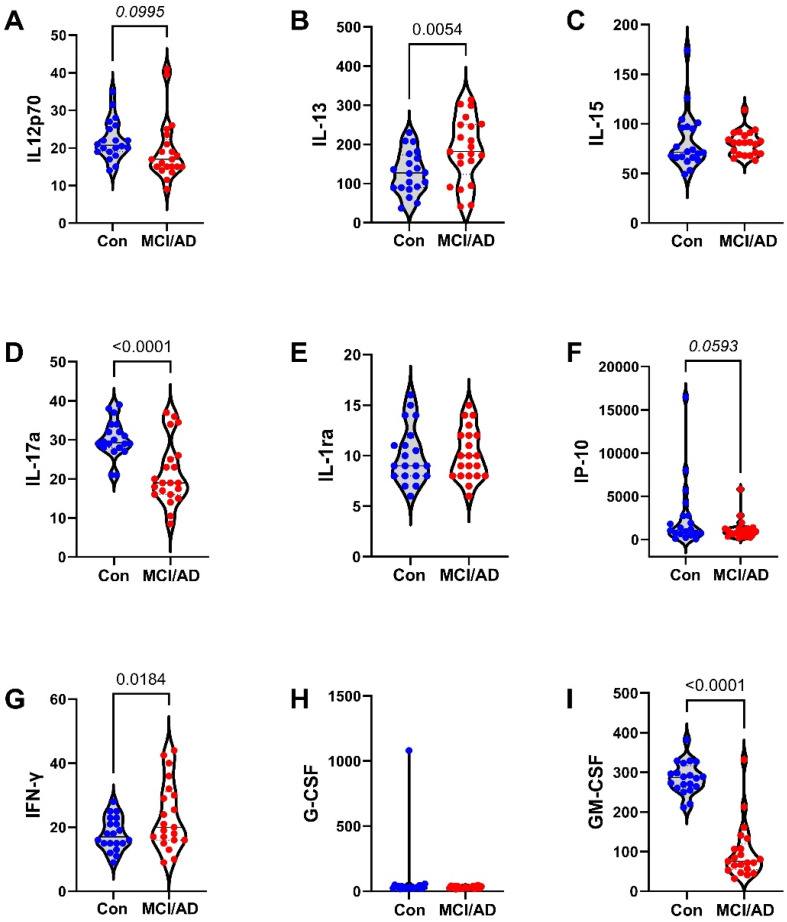
MCI/AD effects on CSF cytokine and chemokine expression—2: Bead-based multiplex ELISAs were used to measure cytokines, chemokines, or trophic factors in CSF. Violin plots depict inter-group comparisons with individual values corresponding to the control (Con; N = 21) and MCI/AD (N = 18) levels of (**A**) IL-12p70, (**B**) IL-13, (**C**) IL-15, (**D**) IL-17a, (**E**) IL-1ra, (**F**) IP-10, (**G**) IFN-γ, (**H**) G-CSF, and (**I**) GM-CSF analyzed using Student’s *t*-test. Significant (*p* < 0.05) and statistical trend (*0.05* ≤ *p* ≤ *0.10*; *italicized*) inter-group differences are shown within each panel.

**Figure 7 biomedicines-11-02394-f007:**
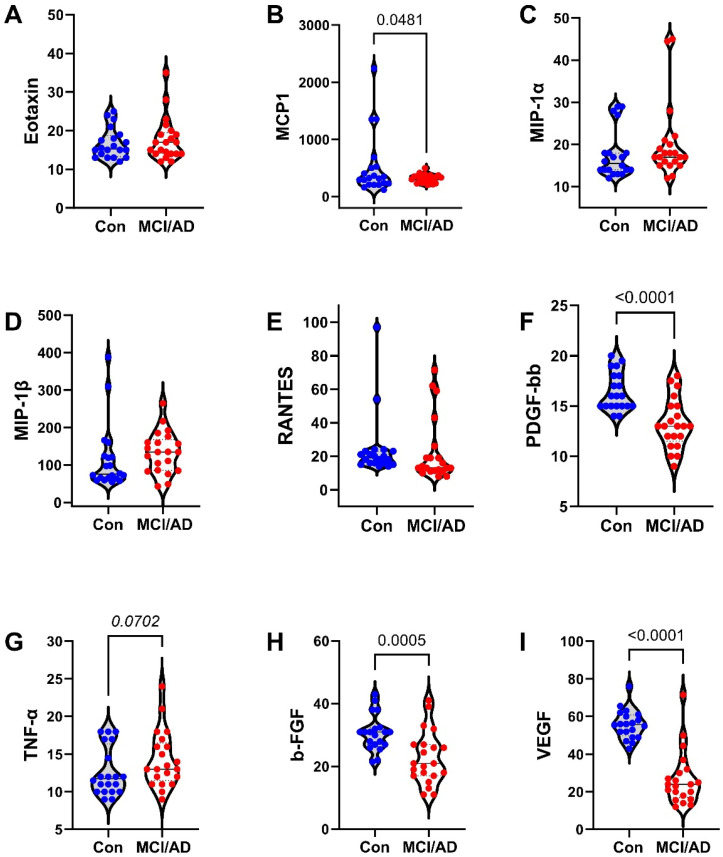
MCI/AD effects on CSF cytokine and chemokine expression—3: Bead-based multiplex ELISAs were used to measure cytokines, chemokines, or trophic factors in CSF. Violin plots depict inter-group comparisons with individual values corresponding to the control (Con; N = 21) and MCI/AD (N = 18) levels of (**A**) eotaxin, (**B**) MCP-1, (**C**) MIP-1α, (**D**) MIP-1β, (**E**) RANTES, (**F**) PDGF-bb, (**G**) TNF-α, (**H**) b-FGF, and (**I**) VEGF analyzed using Student’s *t*-test. Significant (*p* < 0.05) and statistical trend (*0.05* ≤ *p* ≤ *0.10*; *italicized*) inter-group differences are shown within each panel.

**Figure 8 biomedicines-11-02394-f008:**
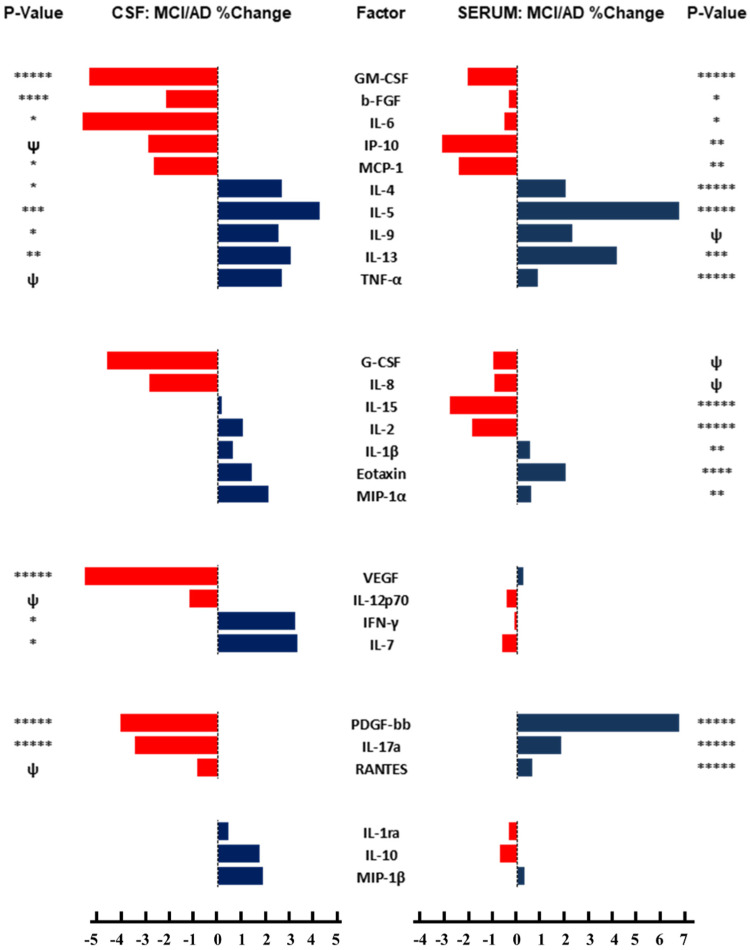
Databar plots depicting the effects of MCI/AD on CSF and serum cytokine expression profiles: Multiplex magnetic-bead-based ELISAs measured 27 human cytokines and chemokines. Databar plots compare the mean percentage changes in immunoreactivity for MCI/AD versus control CSF (left) and serum (right). Reductions in immunoreactivity are indicated by bars to the left of the median, while increases are shown by bars to the right. The scales at the bottom reflect 20% incremental reductions (negative values) or increases (plus values to the right of 0) in cytokine expression in MCI/AD versus control groups. Statistical comparisons were conducted by unpaired *t*-tests: * *p* < 0.05; ** *p* < 0.01; *** *p* < 0.005; **** *p* < 0.001; ***** *p* < 0.0001; ψ 0.05 ≤ *p* ≤ 0.10 (statistical trend).

**Table 1 biomedicines-11-02394-t001:** Cytokines/chemokines—systemic and CNS actions.

Cytokine/Chemokine	Full-Other Names	Systemic Functions	Roles in Neurodegeneration	Citations
b-FGF	Basic fibroblast growth factor; FGF2	Angiogenic and broad-spectrum mitogenic factor; localized in basement membranes and vascular subendothelial extracellular matrix; cytoprotective; role in wound healing	Both a neurotrophin and mediator of neuronal injury; signals through networks involved in neurogenesis (temporal lobe) and post-traumatic injury related neurodegeneration. Increased levels in various neurodegenerative diseases. Immunoreactivity detected in astrocytes, senile plaques, neuropil threads, and neurofibrillary tangles in AD.	[26,27,28,29,30]
Eotaxin	Eosinophil chemotactic protein; CCL11 (eotaxin-1), CCL24 (eotaxin-2), and CCL26 (eotaxin-3)	CC chemokine subfamily of proteins chemotactic for eosinophils; binds to CCR2, CCR3, CCR5; levels elevated in aging; active consumption of cannabis and schizophrenia	Elevated in CSF and plasma of aging mice; impairs neurogenesis, cognition, memory; plasma levels elevated in AD and other forms of neurodegeneration	[26,31,32]
G-CSF	Granulocyte colony stimulating factor; CSF-3	Stimulates granulocyte activation, proliferation, survival, and differentiation, produced by endothelium and macrophages	Supports neuroprotection due to anti-apoptotic effects via pAKT activation	[26,33,34]
GM-CSF	Granulocyte- macrophage colony stimulating factor; CSF-2	Cytokine promotes host defenses; stimulates stem cells to generate granulocytes and induces differentiation of monocytes into macrophages or dendritic cells	Neuroprotective. Prevents neurodegeneration in MPTP models of PD; mediates autoimmune encephalitis	[26,33,35,36,37,38,39,40]
IFN-γ	Interferon-gamma; type II interferon	Pro-inflammatory cytokine and potent activator of macrophages; plays a role in mediating innate and adaptive immune responses; delayed immune response	Mediates delayed post-ischemia neurodegeneration via IFN-γ secreted by splenic macrophages; promotes inflammatory-mediated impairment of neural stem and neuroprogenitor cell maturation and differentiation	[10,41,42,43,44]
IL-10	Interleukin-10; cytokine synthesis inhibitory factor	Anti-inflammatory cytokine; suppresses pro-inflammatory genes and cytokine secretion in macrophages and neutrophil	Neuroprotective; prevents LPS-induce neurodegeneration; expressed in microglia	[26,35,45]
IL-12 (p70)	Interleukin-12; p70 is the active heterodimer	Pro-inflammatory cytokine; promotes antigen expression in B cells, macrophages, neutrophils, and dendritic cells. Bolsters production of IFN-γ and TNF-α; stimulates IL-7 in macrophages	Induces excitotoxic neuronal injury in brain by stimulating IL-7 in microglia	[26,35,45,46,47]
IL-13	Interleukin-13	Cytokine secreted by TH2 T helper cells; effects similar to those of IL-4 but mainly reduces allergic inflammatory responses; reduces TH2 helper cell functions; mediates pro-inflammatory responses such as enhanced secretion of IgE by activated B cells	Potentially neuroprotective for cortical neurons; modulates cortical excitability; expression correlates with Aβ deposition in multiple sclerosis	[48,49]
IL-15	Interleukin-15	Pleiotropic pro-inflammatory cytokine, structurally similar to IL-2, produced by activated monocytes, macrophages, and dendritic cells. Promotes T cell proliferation and cytotoxicity via NK and cytotoxic T cells	Potential biomarker for AD due to elevated serum levels; produced by activated astrocytes	[17,50]
IL-17A	Interleukin-17A	Pro-inflammatory cytokine produced by T helper cells and induced by IL-23. Recruits monocytes and neutrophils to sites of inflammation; role in auto-immune diseases and microbial defenses	T-cell-mediated delayed phase inflammatory injury in ischemic stroke	[51,52,53]
IL-1β	Interleukin-1beta; leukocyte pyrogen; leukocyte activating factor	Pro-inflammatory cytokine produced by activated macrophages; promotes p53-mediated apoptosis	Expressed by microglia in response to injury and exacerbates neuronal injury [IL-1]; causes excitotoxic neurodegeneration via increased generation of glutamate and increases MS progression by way of p53-linked apoptosis; causes death of oligodendrocytes; positive effects include enhanced synaptic transmission	[54,55,56]
IL-1RA	Interleukin-1 receptor antagonist; IL-1 inhibitor	Increases adhesion molecule expression; induces metalloproteinases and prostaglandins	Neuroprotective: inhibits cytotoxic, ischemic, excitotoxic, and traumatic injury in the brain.	[56]
IL-2	Interleukin-2	Cytokine-signaling regulator of activities in leukocytes responsible for immunity; increases T cell proliferation; activates B cells	Neuroprotective for maintaining septal–hippocampal cholinergic neurons; however, high levels cause cognitive dysfunction	[33,57]
IL-4	Interleukin-4	Cytokine induces differentiation of naïve T cells; regulates immune responses, both adaptive and humoral; reduces Th1, IFN-γ, macrophages, and dendritic cell IL-12 via anti-inflammatory actions	May regulate dopaminergic functions in neuron; similar effects as those associated with IL-13.	[26,49,58,59,60]
IL-5	Interleukin-5	Pro-inflammatory cytokine; produced by Th2 T helper cells; promotes activated B cell proliferation, maturation, and immunoglobulin secretion	Induces proliferation and activation of microglia; increases nitrite production and probably nitrosative stress; serum levels elevated in major depressive disorders; mediates its effects on CNS plasticity by utilizing neural-plasticity-related RAS GTPase-extracellular signal-regulated kinase (Ras-ERK) pathway	[61,62]
IL-6	Interleukin-6	Pro-inflammatory cytokine and anti-inflammatory myokine; induces B and T cell proliferation; induces expression of protease inhibitors; macrophages and T cells secreted to enhance immune responses	Expressed in microglia; accumulates around amyloid beta cortical senile plaques; increased levels elevated in PBMCs from AD subjects; MPTP models of PD induce IL-6, but paradoxically neuroprotective.	[18,63,64,65,66,67]
IL-7	Interleukin-7	Hematopoietic growth factor made by stromal, neuronal, dendritic, hepatocellular, and epithelial cells. Positive regulator of B and T cell development and differentiation	CNS and peripherally increased in association with CNS autoimmune diseases (MS/EAE); promoted by elevated levels of TNF-α, IL6, and IFN-γ; increases proliferation of myelin-activated T cells	[26,46,68,69]
IL-8	Interleukin-8	Chemokine ligand (C-X-C motif); regulates neutrophil migration by signaling through CXCR2; induces expression of proinflammatory proteases MMP-2 and MMP-9; induces proapoptotic protein Bim (Bcl-2-interacting mediator of cell death) and cell death	Levels increased by brain injury; higher levels propagate secondary injury	[26,43,54,67,70,71,72]
IL-9	Interleukin-9	Cytokine cellular signaling molecule that modulates pro-inflammatory responses, stimulating proliferation and inhibiting apoptosis; roles in autoimmune disease and asthma	Increased production in AD brain cells. Promotes T cell migration into the CNS	[66,73,74,75]
IP-10	Interferon gamma induced protein 10; CXCL10	Chemokine binds to cell-surface CXCR3 receptors to activate monocyte /macrophage chemoattraction of dendritic cells, NK cells, and T cells. Promotes adhesion of T cells to endothelial cells, antitumor activity, and angiogenesis	Elevated in several neurodegenerative diseases and in MS; mediates stroke-induced neurodegeneration	[41,76,77,78]
MCP-1	Monocyte chemoattractant protein 1; CCL2 (chemokine motif ligand 2)	Chemokine anchored in the plasma membrane and secreted by monocytes, macrophages and dendritic cells, mainly in response to PDGF and CCR2 and CCR4 surface receptors; attracts monocytes	Induced in astrocytes by PDGF-BB; attracts monocytes, promoting their transmigration through a disrupted blood–brain barrier. Increased levels impair attention, executive function, and psychomotor speed.	[26,78,79,80,81,82]
MIP-1α	Macrophage inflammatory protein 1 alpha; chemokine motif ligand 3 (CCL3)	Chemokine with chemoattraction for T cells, NK cells, monocytes, and immature dendritic cells; induces release and synthesis of as IL-1, IL-6 and TNF-α, i.e., pro-inflammatory cytokines from macrophages and fibroblasts.	Promotes neurodegeneration by attracting infiltration of microglia and macrophages; increased expression associated with spongiform neurodegeneration caused by oncornavirus	[18,36,71,80,82,83,84]
MIP-1β	Macrophage inflammatory protein 1 beta; chemokine motif ligand 4 (CCL4)	Chemokine with chemoattraction for NK and T cells with actions similar to MIP-1α. Interacts with CCL3.	Impairs attention, executive function, and psychomotor speed; increased expression with oncornavirus-induced spongiform neurodegeneration	[26,36,71,83]
PDGF-BB	Platelet-derived growth Factor-BB	Chemokine for monocytes and neutrophils; mitogenic for cells of mesenchymal origin; Two forms of PDGF-B dimerization exist: PDGF-BB (homodimer) or PDGF-AB (heterodimer with PDGF-A).	Neuroprotective; promotes neuronal survival; induces neurogenesis in dopaminergic neurons; however, also induces MCP-1 in astrocytes	[26,82,85,86,87,88,89]
RANTES	Regulated upon activation, normal T-cell expressed, and secreted	Chemokine with chemoattraction for T cells and leukocytes, promotes monocyte adhesion to endothelial cells. Binds to CCR1, CCR3, and CCR5 receptors	Major chemokine expressed in brain; including reactive astrocytes in mouse brains infected with scrapie virus; potential for neuroprotection post-ischemic stroke via neuronal induction of neurotrophic factors within peri-infarct zones, leading to enhanced neuronal survival via autocrine or paracrine mechanisms	[90,91,92]
TNF-α	Tumor necrosis factor-alpha; cachexin	Pro-inflammatory cytokine of activated macrophages; binds to TNFR1; induces expression of other cytokines, chemokines (RANTES), metalloproteinases, and adhesion molecules in the setting of acute phase responses; pathogenic role in cachexia, fever (hyperpyrexia), inflammatory responses, and cellular apoptosis. Anti-tumor and anti-viral effects. Dysregulated expression in cancer, psoriasis, and inflammatory bowel disease.	Dysregulated expression in neurodegeneration including AD and in major depression. In neurodegeneration, TNF-α induces neuronal excitotoxic injury (via glutamate); accumulates around senile plaques; induced by MPTP; neuronal excitoxicity; can also increase synaptic transmission	[54,64,65,77,93]
VEGF	Vascular endothelial growth factor; vascular permeability factor	Trophic factor in the PDGF subfamily; stimulates de novo vasculogenesis and angiogenesis, fibroblast proliferation, and monocyte/macrophage migration; restores oxygen supply to tissues injured by deprivation; increases microvascular permeability; levels elevated in diabetes and cancer.	CSF levels elevated in normal brain aging; may be neuroprotective, reduced CNS/CSF levels correlate with hippocampal atrophy and loss of executive functions and memory; interactive effect with Aβ	[18,94,95,96,97,98,99]

Abbreviations (first column) and full names (second column) of the 27 cytokines, chemokines, and trophic factors measured in control-, MCI-, and AD-serum- and CSF-paired samples from the same participants. General systemic sources of factor expression and their functions are listed in column 3, and CNS cellular sources and functions are listed in column 4. In addition to review articles cited at the top of column 5, specific references corresponding to individual cytokines/chemokines are provided.

**Table 2 biomedicines-11-02394-t002:** Subject Demographics.

	Control	MCI	AD
Number Subjects	21	8	10
Age: Years ± S.D [Range]	45.6 ± 11.8 [28–77]	69.1 ± 7.2 [59–77]	67.5 + 11.3 [49–83]
Sex: M/F	11/10	7/1	5/5
MMSE Score ± SD [Range]	N.D.	26.4 ± 3.1 [22–30]	21.9 ± 5.5 [14–28]

Comparisons of mean ages, sex ratios, and MMSE scores among subjects diagnosed as control, MCI, or AD. MMSE scores were not obtained (N.D.) for control subjects.

## Data Availability

Data can be obtained from SMdlM upon request.

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
