# Peer review of "Concordant and Discordant Cerebrospinal Fluid and Plasma Cytokine and Chemokine Responses in Mild Cognitive Impairment and Early-Stage Alzheimer’s Disease"

_biomedicines, 2023, doi:10.3390/biomedicines11092394_

Round 1
Reviewer 1 Report
Ref. biomedicines-2545207
The authors measured several pro-inflammatory and anti-inflammatory/neuroprotective molecules in CSF and serum of patients with MCI or AD and observed moderately greater pro-inflammatory than anti-inflammatory responses in CSF and serum as compared to controls. They also provide evidence that there are distinct neuroinflammatory responses in the CNS of patients with MCI/AD unrelated to systemic mechanisms.
Since neuroinflammation seems to be significant mechanism of neurodegeneration in AD even in the early stages, such studies are welcome since they may offer possible targets for therapeutic interventions especially at the MCI stage.
The study is very interesting and well conducted. However there are some major points:
(a) Were patient groups defined according to clinical criteria alone or, in addition, they were all classified as Alzheimer’s disease or Alzheimer’s continuum according to AT(N)? A clinically-based diagnosis is not always enough. For example MCI is a heterogeneous group. Did all patients with MCI suffer of MCI due to AD (based on CSF biomarkers)? These points should be clarified in the text.
(b) I could not find table 2 (demographics). Also, I was not able to find supplementary tables 1 and 2.
(c) Page 5, subsection Concordant and discordant…: “The data were subdivided into five clusters based on their degrees of CSF-serum concordant directional responses (Figure 4)”. It seems to be Figure 8 rather than Figure 4.
(d) Figures 5-7 are not referred or explained in the Results section
(e) It was observed that controls had lower age than the patients. Then what was the effect of age in the various molecules studied? ANOVA and t-tests are reported. I am sure that analysis of covariance (ANCOVA) was performed including age as a covariate, in order to control for age, but this should be clearly reported by the authors.
Author Response
Also see attached document--same content is included
The authors measured several pro-inflammatory and anti-inflammatory/neuroprotective molecules in CSF and serum of patients with MCI or AD and observed moderately greater pro-inflammatory than anti-inflammatory responses in CSF and serum as compared to controls. They also provide evidence that there are distinct neuroinflammatory responses in the CNS of patients with MCI/AD unrelated to systemic mechanisms.
Since neuroinflammation seems to be significant mechanism of neurodegeneration in AD even in the early stages, such studies are welcome since they may offer possible targets for therapeutic interventions especially at the MCI stage.
The study is very interesting and well conducted. However there are some major points:
- Were patient groups defined according to clinical criteria alone or, in addition, they were all classified as Alzheimer’s disease or Alzheimer’s continuum according to AT(N)? A clinically-based diagnosis is not always enough. For example MCI is a heterogeneous group. Did all patients with MCI suffer of MCI due to AD (based on CSF biomarkers)? These points should be clarified in the text.
At the end of the first paragraph of the Methods-Human Subjects section, we added: “In addition, the MCI patients were followed in the Memory Disorders Center and subsequently (post serum and CSF sampling) determined to have evidence of AD.”
- I could not find table 2 (demographics). Also, I was not able to find supplementary tables 1 and 2.
Our sincere apologies for the omissions. All cited Tables have been included in the revised manuscript.
- Page 5, subsection Concordant and discordant…: “The data were subdivided into five clusters based on their degrees of CSF-serum concordant directional responses (Figure 4)”. It seems to be Figure 8 rather than Figure 4.
Thank you for the correction. Indeed the Figure number is 8 and not 4.
- Figures 5-7 are not referred or explained in the Results section
Thank you for the correction. The figures had been re-numbered but the changes were not correctly noted in the manuscript. All figures are now referenced and linked to results.
- It was observed that controls had lower age than the patients. Then what was the effect of age in the various molecules studied? ANOVA and t-tests are reported. I am sure that analysis of covariance (ANCOVA) was performed including age as a covariate, in order to control for age, but this should be clearly reported by the authors.
We added the following comment to the limitations section of the Discussion: The significant difference in mean age may have contributed to some inter-group differences. However, since 7 of 27 serum and 11 of 27 CSF factors were not significantly modulated by diagnosis, and 10 of 27 factors were discordantly modulated by diagnosis, it is likely that the observed pathophysiological responses were MCI/AD-related rather than strictly age-dependent.

Reviewer 2 Report
The authors investigated neuroinflammation in a mixed cohort of MCI and early AD patients.
Major concerns:
- With 10 AD and 8 MCI patients, the diseased cohort is extremly small and it is questionable if the statistical powers is given considering this sze. To effectivley compare these groups, a higher sample size is advised. This holds particularily true for Fig 2 (ff) where both groups are combined. It would be interesting to understand if MCI as AD precursor demonstrates differences in neuroinflammation, particularily considerng the study question.
- A statistical "trend" as the authors call it, is not a significant result. Trends should not be highlighted as meaningful results. Particularily, as the authors even consider results a trend that are above p = 0.1.
- For many measurements, there are extreme outlies. Specifically, in the healthy group. What is the reason for these outlieres? The results would benefit from min/max exclusion and additional samples.
- For all results: Please display individual datapoints as points. For example as bar plot with each patient as dot.
Minor comments:
- The introduction is quite lengthy and does not convincingly convey the previous knowledge.
The english is fine.
Author Response
Information also included in the attachment
The authors investigated neuroinflammation in a mixed cohort of MCI and early AD patients.
Major concerns:
- With 10 AD and 8 MCI patients, the diseased cohort is extremely small and it is questionable if the statistical powers is given considering this sze. To effectively compare these groups, a higher sample size is advised. This holds particularly true for Fig 2 (ff) where both groups are combined. It would be interesting to understand if MCI as AD precursor demonstrates differences in neuroinflammation, particularily considerng the study question.
The relatively small sample sizes are acknowledged as limitations of the study. However, it is also very difficult to obtain sufficient volumes of paired CSF and serum samples for clinical diagnoses and research in a clinical setting. Supplementary Tables 2 and 3 show how MCI and AD differed with respect to neuroinflammation. Importantly, the trends and responses were similar, justifying their grouping for comparisons with controls.
- A statistical "trend" as the authors call it, is not a significant result. Trends should not be highlighted as meaningful results. Particularily, as the authors even consider results a trend that are above p = 0.1.
We have defined and minimized use of the term ‘statistical trend’ and now focus the inter-group differences that reached statistical significance [P≤ 0.05]. However, 0.05 <P < 0.10 are shown over the relevant graphs as the few relevant trends are briefly mentioned or discussed in the manuscript.
- For many measurements, there are extreme outlies. Specifically, in the healthy group. What is the reason for these outlieres? The results would benefit from min/max exclusion and additional samples.
The reconfigured graphs illustrate the outlier points. It is noteworthy that the relatively few prominent outliers did not impact the statistical comparisons by rendering the inter-group differences statistically significant when the majority of data points overlapped.
- For all results: Please display individual datapoints as points. For example as bar plot with each patient as dot.
The graphs have been reconfigured as suggested. Comparisons were made using Violin plots with superimposed individual datapoints for all factors (Figures 2-7).
Minor comments:
- The introduction is quite lengthy and does not convincingly convey the previous knowledge.
We have edited and shortened the Introduction aiming to convey previous knowledge as suggested.
Comments on the Quality of English Language
The english is fine.

Round 2
Reviewer 1 Report
The authors modified the manuscript according to reviewers’ suggestion and the quality of the paper is now substantially improved. I believe it can be published in the present form.
Author Response
Thank you. No further comments are required.
Reviewer 2 Report
The authors addressed all comments.
Author Response
Thank you. no further comments are required